# A Pattern-Recognizer Artificial Neural Network for the Prediction of New Crescent Visibility in Iraq

**Ziyad T. Allawi**

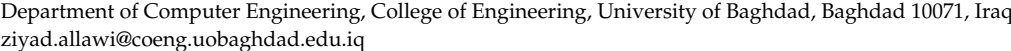

Department of Computer Engineering, College of Engineering, University of Baghdad, Baghdad 10071, Iraq; ziyad.allawi@coeng.uobaghdad.edu.iq

**Abstract:** Various theories have been proposed since in last century to predict the first sighting of a new crescent moon. None of them uses the concept of machine and deep learning to process, interpret and simulate patterns hidden in databases. Many of these theories use interpolation and extrapolation techniques to identify sighting regions through such data. In this study, a pattern recognizer artificial neural network was trained to distinguish between visibility regions. Essential parameters of crescent moon sighting were collected from moon sight datasets and used to build an intelligent system of pattern recognition to predict the crescent sight conditions. The proposed ANN learned the datasets with an accuracy of more than 72% in comparison to the actual observational results. ANN simulation gives a clear insight into three crescent moon visibility regions: invisible (I), probably visible (P), and certainly visible (V). The proposed ANN is suitable for building lunar calendars, so it was used to build a four-year calendar on the horizon of Baghdad. The built calendar was compared with the official Hijri calendar in Iraq.

**Keywords:** pattern recognition; artificial neural network; deep learning; crescent moon early sighting

## 1. Introduction

The early sighting of a new crescent moon is one of the earliest astronomical activities performed by human civilizations. This sighting marked the start of the lunar month in many such civilizations. The first evidence of these activities came to us from the Babylonians (6th Century B.C.), who lived in the fertile lands of Mesopotamia (modern-day Iraq) [1]. They relied on the crescent moon to indicate the start of their calendar months and years. They used a standard criterion to predict the moon's sight, which was sufficient for that purpose. Other civilizations, such as Indians and Chinese, also used lunar sight and still do to this day, although use different criteria to determine the start of a lunar month [2]. Jewish people also still use the moon cycle to indicate the start of the months in their lunisolar calendar, although they use arithmetic calculation rather than moon sighting.

Centuries after the Babylonians, in the 7th Century C.E., a new religion, Islam, had emerged from the deserts of the Arabian Peninsula. Three of the five main pillars of Islam depend on the start of the lunar months. They are Fasting-Breakfasting "*Siam-Fitr*", Pilgrimage "*Hajj*", and Alms "*Zakat*". The first two pillars are performed in a specific lunar month in a year, while the third is performed once in the first month of each lunar year [2]. Reckoning of the first crescent moon after the conjunction is an essential indication for Muslims to begin their lunar months and perform their religious duties.

Through the Islamic golden age (between the 9th and 11th centuries C.E.), many famous Muslim astronomers laid down specific criteria for moon sighting that depended on the age of the moon (the time passed since the conjunction) and the lag of the moon (the time between the sunset and the moonset) [3]. Some considered the arc length of celestial degrees between the sun and the moon (elongation) at sunset. However, they could not predict the sighting perfectly because they did not consider the local conditions of the horizon at the time of observation, and they omitted the effect of changing the apparent width of the crescent moon and its relation with the brightness of the sky [4].

Since the start of the previous century, various theories have been proposed to design a criterion for moon visibility suitable to predict the start of lunar months. These theories relied on drawing a borderline between positive (visible) and negative (invisible) observations or on defining a visibility region using interpolations. All these methods were empirical and depended on the subjective viewpoint of the observer. Many observation records yielded incorrect predictions [5]. There is also interference between the two mentioned regions, which was difficult separate empirically. This situation could be resolved by drawing a barrier region between them. Therefore, an objective-oriented method that depends on the presented datasets themselves should make the visibility regions more realistic and reliable.

The development of machine learning (ML) techniques in recent years resulted in many considerable advancements in modern technology. ML algorithms have proven their ability in numerous fields, such as pattern recognition. *Artificial neural networks* (ANN's) are of the most widely used technology in pattern recognition due to their ability to analyze and learn patterns hidden inside the data that they consider, as well as their ability to train themselves on new data to rectify and improve their performance. They can predict the output of any input not presented in the training data, making them flexible in handling lost and misleading information. Many studies have been performed related to ML with ANNs in the field of pattern recognition and classification [6–9].

In this study, a two-layered ANN was used to learn moon sighting data tables provided by Schaefer, Yallop, and Odeh to generate a moon sighting region recognizer to predict the possible condition of the first sighting of a new crescent moon. This ANN was used to build a calendar for four Islamic Hijri years (1440–1443 A.H.) based on the local horizon of Baghdad, the capital of Iraq, as a part of a larger project to build a unified Islamic Hijri Calendar for Iraq. The results of the ANN recognizer were compared with the current official Hijri Calendar in Iraq, which relies on astronomical calculations rather than moon sighting.

The remainder of this study is organized as follows. In Section 2, we outline a literature review of related works. In Section 3, we illustrate the methods used to calculate the essential parameters for moon sighting and demonstrate the observational dataset and the design of the proposed pattern recognizer ANN. In Section 4, we present the results of ANN training and the newly constructed calendar. In Section 5, we discuss the obtained results.

## 2. Related Works

Many astronomers and scientists developed arithmetic criteria for the first visibility of the crescent moon at the beginning of the previous century. Many of their criteria were based on the minimum elongation of the moon suitable for visibility.

Developing accurate criteria for moon sighting in the modern era began with Fotheringham [10] in 1910 and Maunder [11] in 1911, who categorized moon sighting regions as visible/invisible using moon observation data collected by Julius Schmidt in Athens, Greece, between 1859 and 1867. They used one parameter for crescent visibility, i.e., elongation, which is the distance in arc degrees between the sun and the moon. They claimed that the shortest elongation for naked-eye visibility should be 12° (Fotheringham) or 11° (Maunder). In 1932, Danjon [12] claimed that the moon could not be seen if its elongation is less than 7°, thereafter referred to in the literature as "Danjon's limit". Many moonsighting campaigns led by McNally [13] and Schaefer [14] attempted to break that limit [15]; the results were recorded in tables and used by many others, such as Yallop [16] and Odeh [17].

McNally claimed that the moon could reflect the light from the sun to the earth if its elongation was 5.5° at least. However, Schaefer criticized McNally's claim and confirmed Danjon's limit [18]. Ilyas [19] agreed with Maunder's limit and modified it to 10.5° for naked-eye visibility. Yallop [16] agreed with Ilyas' limit, although decreased it to 10°. Fatoohi [1] modified Danjon's limit to 7.5° after considering many data records which belong to the Babylonian era. Finally, Odeh [17] claimed that one of the recorded observations by the *Islamic Crescent Observation Project* (ICOP) broke the Danjon's limit. In that

observation, the crescent was seen by a telescope at elongation of 6.4°. However, this observation is unique and unreliable without further investigations and similar observational conditions [20].

Table 1 illustrates the minimum elongation for visibility according to several authors:

**Table 1.** List of the minimum values of moon elongation for early visibility according to several authors.

| Research Author | Minimum Value of Elongation |
|---|---|
| Fotheringham [10] | 12° |
| Maunder [11] | 11° |
| Ilyas [19] | 10.5° |
| Yallop [16] | 10° |
| Fatoohi [1] | 7.5° |
| Danjon [12] and Schaefer [21] | 7° |
| Odeh [17] | 6.4° |
| McNally [13] | 5.5° |

The criteria above are simple because they give one parameter only to judge whether the crescent would be seen or not. However, ancient and contemporary astronomers used many complicated criteria. A comprehensive list of these criteria and their parameters is shown in [3].

Many previous authors attempted to add other parameters to moon visibility criteria. They considered the physical aspects and their relation with early visibility, such as sky brightness and eye perception [21]. However, other astronomers omitted the effect of the atmosphere for theoretical visibility [13].

Fotheringham-Maunder proposed the first mathematical criterion, which extracted it from observational data [10,11]. It has two parameters. These parameters are the *azimuthal difference* between the sun and the moon (which is abbreviated as $DAZ$) and the altitude difference (which is referred to as *the arc of vision* and abbreviated as $ARCV$) between the sun and the moon. These parameters are computed at the time of sunset at the observation location. This criterion draws a line between the positive observations (visible by the naked eye V) and negative observations (invisible by the naked eye I). For instance, if the moon's position was under that line, it would not be seen, and vice versa.

In 1977, Bruin [4] modified Fotheringham-Maunder's criterion. He considered the western sky, the moon's surface brightness, and the solar depression (the altitude of the sun below the horizon). Bruin presented a new parameter, (*the crescent width*, abbreviated as $w$). The crescent width is the arclength of the crescent, which is proportional to the radius and the illuminated fraction of the moon. Bruin used the arc of vision as a function of the crescent width and found that the moon will not be visible if its width is less than $0.5'$ (30 arcseconds).

Ilyas [22], on the other hand, reformed Bruin's criterion and found that the minimum limit of visible width of the moon was $0.25'$; (15 arcseconds). In [19], he developed a new criterion and claimed that the moon could not be visible if it had been found on the horizon when the depression of the sun was 4° ($DEP = 4°$). He verified the Maunder and Bruin visibility criteria and extended their curves using extrapolation to cover visibility conditions in higher geographic latitudes. Furthermore, Ilyas found that Danjon's limit was just an extrapolation, therefore he increased the limit to 10.5° for naked-eye visibility. This issue was due to observation in the lower-tropical geographic latitudes rather than the middle latitudes when Danjon recorded his observations in Strasbourg Observatory, France.

Schaefer [21] claimed that some of Bruin's data were misleading, and some of his assumptions were incorrect regarding twilight sky brightness, lunar surface brightness, and physiological data of lunar vision. Schaefer added other parameters, such as the visual extinction coefficient and atmospheric clarity. In his study above, he demonstrated the condition of the first moon visibility and claimed that his result was better than the results of Danjon and Bruin.

Yallop [16] used the results of Maunder and Bruin and the observational data collected by Schaefer to derive a new *q*-test criterion based on a published formula in the *Indian Astronomical Almanac* in 1979. That formula was useful for heliacal rising and setting. He presented the concept of the *best time*, which is the time of the optimal observational condition. He claimed that the best time $T_b$ is the time which comes after the sunset $T_s$ in exactly four-ninths of the moon lag, i.e.,:

$$T_b = T_s + \frac{4}{9} Lag \tag{1}$$

where *Lag* is the time period between the sunset and the moonset.

Yallop generated the observational data provided by Schaefer at his best time, then he used the arc of vision *ARCV* (in degrees) versus crescent width *w* (in arcminutes) to develop his criterion as follows:

$$q_Y = \left( ARCV - 11.8371 + 6.3226w - 0.7319w^2 + 0.1018w^3 \right)/10 \tag{2}$$

Table 2 illustrates the crescent conditions according to Yallop:

**Table 2.** Crescent Conditions for Yallop's criterion [16].

| *q* Value | Crescent Condition |
|---|---|
| $q_Y > 0.216$ | Easily visible |
| $0.216 \geq q_Y > -0.014$ | Conditionally visible |
| $-0.014 \geq q_Y > -0.16$ | May need optical aid |
| $-0.16 \geq q_Y > -0.232$ | Should need optical aid |
| $-0.232 \geq q_Y$ | Invisible |

In 2001, Caldwell and Laney [23], a team of two astronomers in the *South African Astronomical Observatory* (SAAO), developed the *SAAO Criterion*, another criterion for crescent moon visibility. This criterion depends on *the arc of light* (abbreviated as *ARCL*) which is slightly less than the elongation, and the arc of vision *ARCV* (for the bright limb of the moon rather than its center) at the time of the sunset. This criterion was simple; it does not need a complicated polynomial to calculate. The SAAO q-test criterion states that:

$$q_S = ARCV + \frac{1}{3} ARCL \tag{3}$$

Table 3 illustrates the crescent conditions according to the SAAO team:

**Table 3.** Crescent Conditions for SAAO criterion [23].

| *q* Value | Crescent Condition |
|---|---|
| $q_S > 11$ | Easily visible |
| $11 \geq q_S > 9$ | Needs optical aid |
| $9 \geq q_S$ | Invisible |

Caldwell [24] investigated the correlation of moon lag with the arc of light. Meanwhile, Qureshi [25] proposed a modification to Yallop's criterion depending on recently collected observation data.

In 2005, Odeh [17] proposed the most recent moon sight criterion. He used the dataset of moon sightings provided by Schaefer and Yallop and added other collected records from many crescent watchers and the ICOP. Odeh used more than 700 records in his study and built his criterion based on the Indian formula by altering Yallop's criterion offset and region control values. Furthermore, he used Yallop's best time for data collection and used *ARCV* and *w* to develop it as follows:

$$q_O = ARCV - 7.1651 + 6.3226w - 0.7319w^2 + 0.1018w^3 \tag{4}$$

Table 4 illustrates the crescent conditions according to Odeh:

**Table 4.** Crescent Conditions for Odeh criterion [17].

| *q* Value | Crescent Condition |
|---|---|
| $q_O > 5.65$ | Easily visible |
| $5.65 \geq q_O > 2$ | May need optical aid |
| $2 \geq q_O > -0.96$ | Needs optical aid |
| $-0.96 \geq q_O$ | Invisible |

Other researchers attempted to formulate visibility criteria for the crescent moon. Some of them proposed a line referred to as the International Date Line (IDL), a line that divides the globe into two lunar month-start regions. They claimed that the line would resolve the problem of differences in lunar months towards constructing a universal Hijri calendar [26,27]. Al-Mostafa [28] proposed a criterion for moon visibility and reckoning the first day of the lunar month in KSA. A recently collected dataset by Alrefay et al. [20] of moon sighting observations is useful for a new robust criterion.

All the previous proposed criteria relied on empirical techniques such as interpolating data of the moon at the observation time. This study aims to divide the moon visibility regions according to a prediction algorithm based on ML using the concept of Pattern Recognition built in the ANN. Before getting to the design procedure, it is necessary to demonstrate the essential parameters of the moon sighting and how to compute them.

## 3. Methodology

### 3.1. Moon Sighting Parameters

The Islamic day (and hence the Islamic lunar month) starts at sunset rather than midnight when the civil standard day starts. Observation of the new crescent moon is due on the time of sunset of the 29th day of the elapsed lunar month. The following day, which starts at that sunset, will be either the 30th day of the current lunar month or the 1st day of the next lunar one. The first visibility of the crescent moon decides the choice between these two cases. If the crescent is visible, the next day will be the 1st day of the next month, or else (the crescent is not visible), the next day will be the 30th day of the elapsed month, and the following day will be the 1st day of the following month.

Figure 1 illustrates the essential parameters of moon sighting:

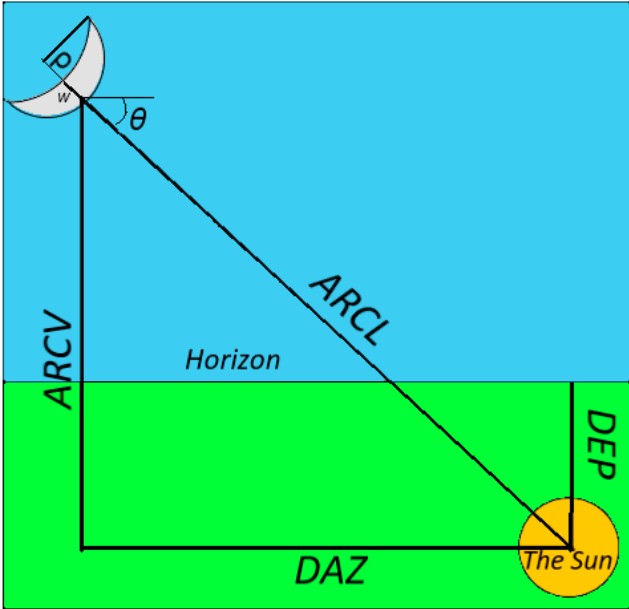

**Figure 1.** Horizontal Coordinates of the Bright Limb of the Crescent Moon Relative to the Sun, along with other parameters.

At the best time (whether it is either the time of sunset when the upper limb of the sun coincides with the horizon or the center of the sun is located below the horizon in several degrees), five parameters are calculated: the equatorial celestial positions of the sun and the moon (declinations and right ascensions), and the apparent radius of the moon disk $\rho_m$.

In this study, the best time was used according to Ilyas [19], when the sun depression is $4°$ ($DEP = 4°$).

The equatorial celestial coordinates of the moon were corrected for parallax as *topocentric* coordinates rather than *geocentric* ones. The astronomical algorithms for calculating such coordinates are available in Jean Meeus's book [29].

The elongation $\psi$ between the centers of the luminaries is calculated as follows:

$$\psi = \arccos\left(\sin\acute{\delta}_m \sin\delta_s + \cos\acute{\delta}_m \cos\delta_s \cos(\acute{\alpha}_m - \alpha_s)\right) \tag{5}$$

where $\acute{\delta}_m$, $\acute{\alpha}_m$ are the corrected equatorial celestial coordinates (declination and right ascension) of the moon,, and $\delta_s, \alpha_s$ are the celestial coordinates (declination and right ascension) of the sun.

The first parameter of the moon sighting, the width of the crescent (abbreviated by $w$), is calculated in terms of arcminutes:

$$w = 60(1 - \cos\psi)\rho_m \tag{6}$$

The orientation angle of the moon's bright limb $\theta$ is calculated from the difference between two arguments:

$$\theta = \arctan\left(\frac{\sin H_m}{\cos\acute{\delta}_m \tan\varphi - \sin\acute{\delta}_m \cos H_m}\right) - \arctan\left(\frac{\cos\acute{\delta}_m \tan\delta_s - \sin\acute{\delta}_m \cos(\acute{\alpha}_m - \alpha_s)}{\sin(\acute{\alpha}_m - \alpha_s)}\right) \tag{7}$$

where $H_m$ is the hour-angle of the moon and $\varphi$ is the geographic latitude [29].

The arc of light $ARCL$, which is the distance between the center of the sun and the bright limb of the moon, is equal to the elongation minus the radius of the moon's disk:

$$ARCL = \psi - \rho_m \tag{8}$$

The other parameters, $ARCV$ and $DAZ$ are calculated in terms of $ARCL$ and $\theta$:

$$ARCV = \arctan(\sin\theta \tan ARCL) \tag{9}$$

$$DAZ = |\arcsin(\cos\theta \sin ARCL)| \tag{10}$$

The values of $w$, $ARCV$ and $DAZ$ are the input to the pattern recognizer ANN which will be trained using the moon sighting dataset.

### 3.2. Pattern Recognizer ANN Architecture

The architecture of the pattern recognizer ANN is shown in Figure 2.

This ANN has three layers: the first one is the input layer (surrounded by the red rectangle) and has three neurons. This layer receives the input parameters ($w$, $ARCV$ and $DAZ$) from the observational dataset. It has an input-to-hidden bias vector ($bi$). The input values of the first neuron are between 0 to 2. In the second and third neurons, the input values are between 0 to 20.

The hidden layer contains five neurons (surrounded by the green rectangle), which receive the input values after scaling them by the input-to-hidden weight matrix ($Wi$) along with the input-to-hidden bias vector. The activation function of the hidden layer is the $\tan\text{sig}()$ function, which is indicated in the figure by the letter (T). This layer has a hidden-to-output bias vector ($bh$).

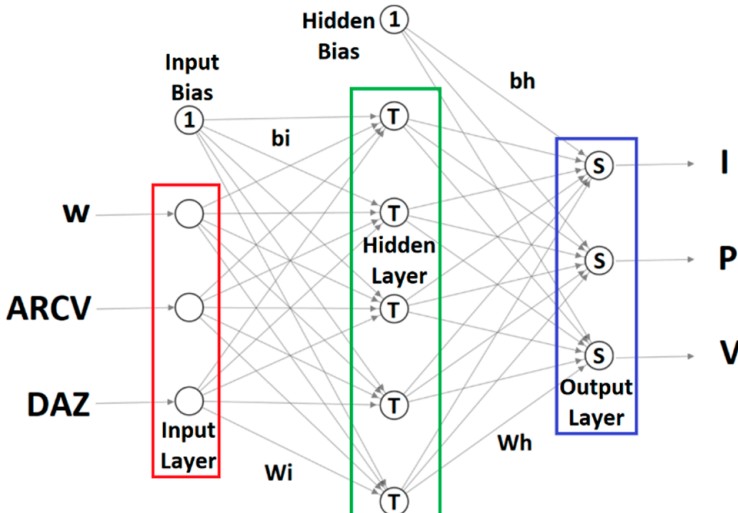

**Figure 2.** Pattern recognizer ANN architecture.

The output layer has three neurons (surrounded by the blue rectangle). Each output neuron represents one of the moon sighting regions (invisible I, probably visible P, and certainly visible V). It receives the results of the hidden neurons after scaling them by the hidden-to-output weight matrix (*Wh*) along with the hidden-to-output bias vector. The activation function of the output layer is the softmax() function, which is indicated in the figure by the letter (S). If any of the output neurons is unity or near unity, its corresponding output will refer to the observation result. The formulae of the two activation functions are illustrated in Equations (11) and (12).

$$y = \tan \text{sig}(x) = \frac{1 - e^{-2x}}{1 + e^{-2x}} \tag{11}$$

$$y_i = \text{softmax}(\mathbf{x}_{n \times 1}, i) = \frac{e^{x_i}}{\sum_{k=1}^{n} e^{x_k}} \tag{12}$$

where $\mathbf{x}_{n \times 1}$ is the vector of the output layer. In this study, $n = 3$ to represent the number of the output values.

The ANN was trained using the *Bayesian Regularization backpropagation algorithm* (BRA), one of the widely used ANN training algorithms for challenging data. BRA updates the weight and bias values according to the Levenberg-Marquardt optimization (LMA). It minimizes a combination of squared errors and ANN weights to determine the correct combination and produce a network that simulates the datasets well. This algorithm uses an adaptive learning rate between 0.005 to $10^{10}$ [30].

Choosing the number of hidden neurons was a tradeoff between the complexity of the ANN (and hence the computation time) and the speed of ANN learning of the observation dataset. If the number of hidden neurons was decreased, the ANN would need more time or additional data to learn the pattern of the dataset efficiently.

*3.3. Training Dataset*

The parameters of the moon's sight were calculated on the day of the observation. The generated data are the Best time according to Ilyas [19], age of the moon *Age*, crescent width *w*, *ARCL*, *ARCV*, and *DAZ*. The parameters were generated from the observational data (Sight Date, Longitude, and Latitude), along with the recorded observational conditions of the crescent moon (Invisible, visible by Telescope, visible by Binoculars, easily Visible, etc.) These targets are used in ANN learning.

The dataset contains 578 entries, each entry for one observation. After combining the dataset of the three authors, several observations were repeated; therefore, it was significant to unify all of them. Furthermore, all morning observations (which were held

before sunrise on the last day of the month) were skipped from the dataset. ANN training was implemented on the evening observations only. Table 5 shows a sample of the dataset. The complete dataset table is found in the Supplementary Materials, Table S1.

**Table 5.** A sample of the Observation Dataset provided by Schaefer, Yallop and Odeh for moon sighting. The sighting parameters are calculated for the "best time" according to Ilyas.

| C | H. Y. | M. | Sight Date | Best Time (UT) | Lat | Lon | Age | W | ARCL | ARCV | DAZ | SO | A |
|---|---|---|---|---|---|---|---|---|---|---|---|---|---|
| O | 1422 | 12 | 12 February 2002 | 16:30 | 43.9 | 18.4 | 8.8 | 0.08 | 5.58 | 0.85 | 5.51 | I | I |
| S | 1408 | 2 | 23 September 1987 | 23:48 | 37.2 | −84.1 | 20.65 | 0.22 | 9.39 | 3.29 | 8.8 | I | I |
| O | 1423 | 2 | 13 April 2002 | 19:20 | 30.5 | −9.7 | 23.96 | 0.26 | 10.57 | 9.14 | 5.33 | I(T) | P |
| S | 1361 | 12 | 8 December 1942 | 21:48 | 40.7 | −74 | 19.8 | 0.34 | 11.32 | 9.78 | 5.74 | V(F) | P |
| O | 1423 | 6 | 9 August 2002 | 17:55 | 10.3 | 9.8 | 22.65 | 0.41 | 12.57 | 12.4 | 2.04 | V | V |
| S | 1393 | 2 | 5 March 1973 | 23:53 | 40 | −85 | 23.75 | 0.38 | 12.24 | 12.2 | 0.97 | V(V) | V |

There are 14 columns in this table. The 1st one represents the catalog origin: (S) for Schaefer; (O) for Odeh. The second column represents the Hijri year in which the observation attempts to recognize its start. The number in the third column represents the number of the lunar month which begins from 1 as Muharram through 12 as Thu-Al-Hijja. The 4th and 5th columns represent the Gregorian calendar date and time. The time is in terms of the coordinated Universal Time (UT) of the observation day. The 6th and 7th columns represent the geographical coordinates of observation (Lat for geographical latitude and Lon for geographic longitude). The 8th column contains the moon's age in hours from the geocentric conjunction until the best time, whereas the 9th column contains the width of the crescent. The 10th, 11th, and 12th columns hold the horizontal coordinates of the bright limb of the crescent moon. These coordinates are relative to the sun in terms of the arc degrees. The 13th column contains the observational results as recorded by Schaefer and Odeh. The meaning of these data values is illustrated in [16,17]. The 14th column contains the observational results as will be interpreted by the pattern recognizer ANN (I as invisible, P as probably visible, and V as certainly visible).

The parameters in columns 9, 11, and 12 were used to train the ANN to fit the output in column 14. Before ANN training, the dataset was distributed on a 2D chart ($ARCV$ vs. $w$) in Figure 3. Every dot represents a single observation. Its color indicates its observation result (red as invisible, green as probably visible, blue as certainly visible).

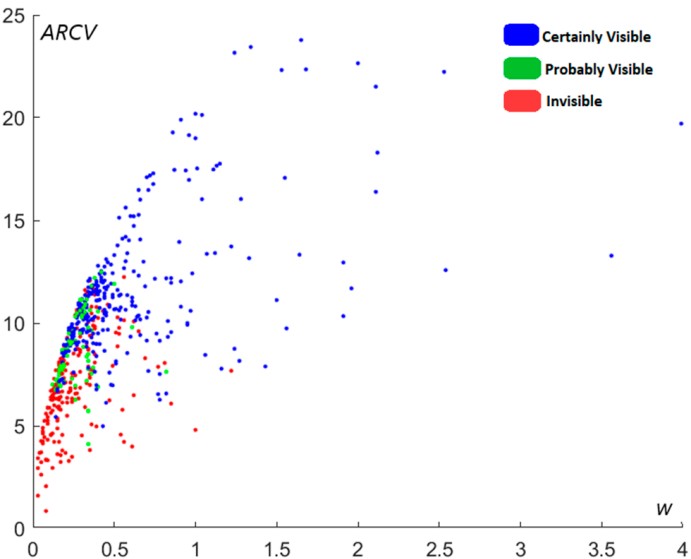

**Figure 3.** Distribution of the training dataset on a chart ($ARCV$ vs. $w$).

## 4. Results

### 4.1. ANN Training

The pattern recognizer ANN was trained by the dataset, which contains 578 entries using MATLAB® R2020b. The training performance MSE was 0.1134 after about 360 training epochs throughout the overall process. Figure 4 illustrates the results of the training.

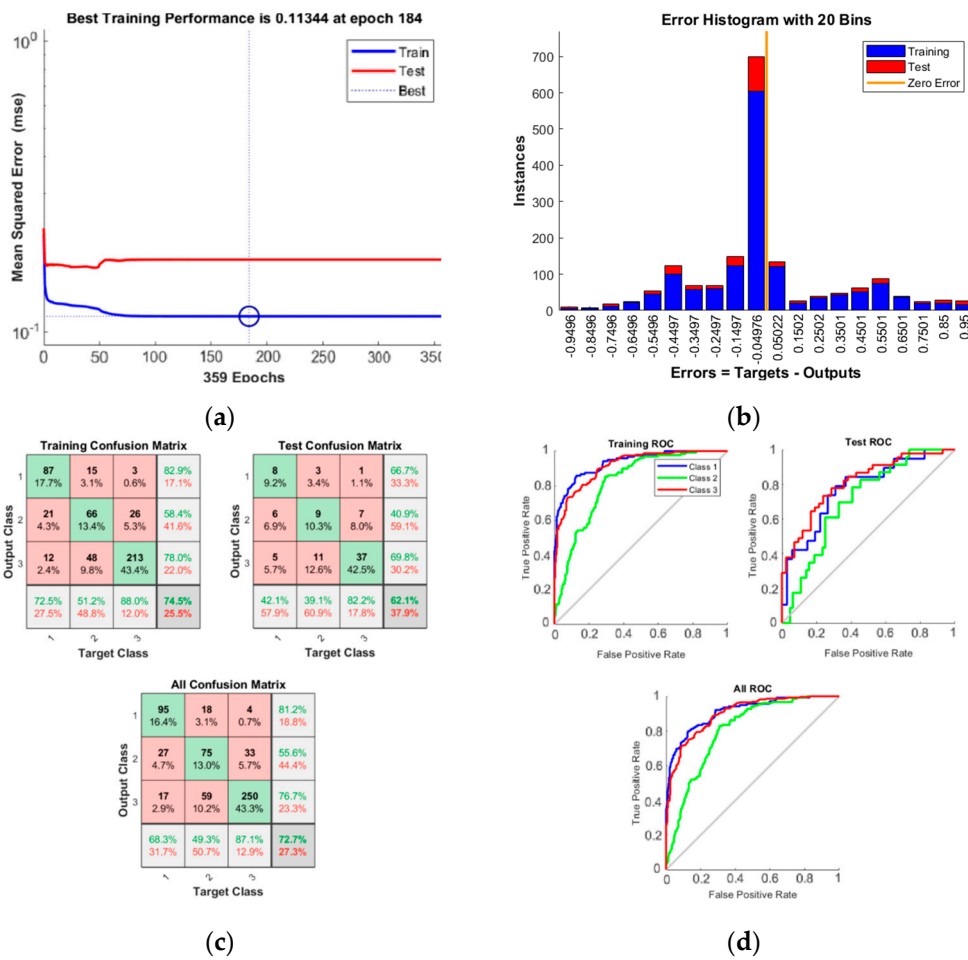

(a)

(b)

(c)

(d)

**Figure 4.** Results of the pattern recognizer ANN training: (**a**) Best training performance with a MSE of 0.1134; (**b**) Error histogram of the dataset entries; (**c**) Confusion table of ANN output relative to the desired target; (**d**) Receiver Operating Characteristic of the ANN.

The results of the training show that the pattern recognizer managed to learn the observation dataset in a high percentage (72.7%). This percentage represents eight identical observations out of eleven.

Figure 4a shows the convergence progress of the ANN. The error reached its minimum at epoch No. 184. The error between the desired outputs (Targets) and actual ANN outputs (Outputs) did not exceed 5% for most of the ANN output values compared with their corresponding desired target values. Figure 4b illustrate that. In Figure 4c, the confusion tables of the results show that 420 observations were true (the three green diagonal squares), whereas fifty-five observations were overestimated, i.e., they should be hardly visible or might be invisible even though they were recorded as visible (the three upper reddish squares). The remaining observations (103) were underestimated, i.e., they should be visible even though they were recorded as invisible (the three lower reddish squares). In Figure 4d, it is clear that the ANN successfully learned the invisible and certainly visible regions (the red and blue lines). However, the ANN learned the region of the probably visible slightly worse (the green line) because it was closer to the center line than the others.

A test dataset was generated to test the pattern recognizer ANN efficiently. The values of the crescent width are between $0.01'$ and $4'$ in a step of $0.01'$. The values of the orientation angle of the moon's bright limb $\theta$ are between $0°$ and $90°$ in a step of $1°$. The value of $ARCL$, $ARCV$ and $DAZ$ are calculated accordingly using $w$ and $\theta$ as follows:

$$ARCL = 20.557\sqrt{w} - 0.259 \tag{13}$$

$$ARCV = \arcsin(\sin\theta \sin ARCL) \tag{14}$$

$$DAZ = |\arctan(\cos\theta \tan ARCL)| \tag{15}$$

The test data contain 36,400 entries. Every entry is sent to the input of the ANN. The output of the ANN was illustrated on a chart according to $ARCV$ vs. $w$. Their colors represent the observation results (red for invisible, green for probably visible, and blue for certainly visible).

### 4.2. Building a Sample Hijri Calendar for Iraq

The theoretical observational site was at a reference place in Baghdad, the capital of Iraq. A famous landmark was chosen, which is the historical Qishla clocktower at the left (eastern) bank of the Tigris river ($Lat = 33.34$; $Lon = 44.387$). The pattern recognizer ANN generated the information for the lunar month start and compared it with the official lunar month start in Iraq mentioned on the ICOP website [31]. Table 6 shows the first days for all the months in the Hijri years (1440–1443):

**Table 6.** The first days of the Hijri lunar months of the years (1440–1443) in astronomical (1) and observational (2) methods. The difference appear only when the moon is "Invisible" on sight day.

| H. Y. | M. | Sight Day | w | ARCV | DAZ | Pred. | First Day (1) | Days | First Day (2) | Days |
|-------|-----|-----------|------|------|------|-------|---------------|------|---------------|------|
| 1440 | 1 | 10 September 2018 | 0.37 | 9.08 | 7.73 | P | 11 September 2018 | 29 | 11 September 2018 | 30 |
| | 2 | 9 October 2018 | 0.11 | 6.26 | 2.1 | I | 10 October 2018 | 30 | 11 October 2018 | 29 |
| | 3 | 8 November 2018 | 0.3 | 9.54 | 5.73 | P | 9 November 2018 | 29 | 9 November 2018 | 30 |
| | 4 | 7 December 2018 | 0.03 | 3.43 | 0.27 | I | 8 December 2018 | 30 | 9 December 2018 | 30 |
| | 5 | 6 January 2019 | 0.06 | 4.34 | 2.24 | I | 7 January 2019 | 30 | 8 January 2019 | 30 |
| | 6 | 5 February 2019 | 0.13 | 5.93 | 4.42 | I | 6 February 2019 | 30 | 7 February 2019 | 29 |
| | 7 | 7 March 2019 | 0.26 | 8.58 | 6.19 | P | 8 March 2019 | 29 | 8 March 2019 | 30 |
| | 8 | 5 April 2019 | 0.07 | 1.25 | 5.28 | I | 6 April 2019 | 30 | 7 April 2019 | 30 |
| | 9 | 5 May 2019 | 0.19 | 5.75 | 6.64 | I | 6 May 2019 | 29 | 7 May 2019 | 29 |
| | 10 | 3 June 2019 | 0.04 | 0.5 | 3.83 | I | 4 June 2019 | 30 | 5 June 2019 | 29 |
| | 11 | 3 July 2019 | 0.32 | 7.88 | 7.69 | P | 4 July 2019 | 29 | 4 July 2019 | 30 |
| | 12 | 1 August 2019 | 0.13 | 5.51 | 4.3 | I | 2 August 2019 | 29 | 3 August 2019 | 29 |
| 1441 | 1 | 30 August 2019 | 0.04 | 3.83 | 0.26 | I | 31 August 2019 | 30 | 1 September 2019 | 29 |
| | 2 | 29 September 2019 | 0.38 | 9.71 | 7.22 | P | 30 September 2019 | 29 | 30 September 2019 | 30 |
| | 3 | 28 October 2019 | 0.11 | 5.97 | 2.21 | I | 29 October 2019 | 30 | 30 October 2019 | 29 |
| | 4 | 27 November 2019 | 0.33 | 9 | 7.17 | P | 28 November 2019 | 29 | 28 November 2019 | 30 |
| | 5 | 26 December 2019 | 0.03 | 2.69 | 2.36 | I | 27 December 2019 | 30 | 28 December 2019 | 30 |
| | 6 | 25 January 2020 | 0.14 | 5.5 | 5.38 | I | 26 January 2020 | 30 | 27 January 2020 | 29 |
| | 7 | 24 February 2020 | 0.28 | 8.55 | 6.68 | P | 25 February 2020 | 29 | 25 February 2020 | 30 |
| | 8 | 24 March 2020 | 0.07 | 0.91 | 5.13 | I | 25 March 2020 | 30 | 26 March 2020 | 30 |
| | 9 | 23 April 2020 | 0.1 | 3.86 | 5.23 | I | 24 April 2020 | 30 | 25 April 2020 | 29 |
| | 10 | 23 May 2020 | 0.24 | 7.91 | 5.96 | P | 24 May 2020 | 29 | 24 May 2020 | 30 |
| | 11 | 21 June 2020 | 0.04 | 3.06 | 2.49 | I | 22 June 2020 | 30 | 23 June 2020 | 29 |
| | 12 | 21 July 2020 | 0.35 | 9.49 | 6.91 | P | 22 July 2020 | 29 | 22 July 2020 | 30 |
| 1442 | 1 | 19 August 2020 | 0.16 | 6.89 | 3.39 | I | 20 August 2020 | 29 | 21 August 2020 | 29 |
| | 2 | 17 September 2020 | 0.05 | 4.29 | 0.55 | I | 18 September 2020 | 30 | 19 September 2020 | 29 |
| | 3 | 17 October 2020 | 0.32 | 8.39 | 7 | P | 18 October 2020 | 29 | 18 October 2020 | 30 |
| | 4 | 15 November 2020 | 0.06 | 3.75 | 2.59 | I | 16 November 2020 | 30 | 17 November 2020 | 29 |
| | 5 | 15 December 2020 | 0.34 | 7.78 | 8.46 | P | 16 December 2020 | 29 | 16 December 2020 | 30 |
| | 6 | 13 January 2021 | 0.08 | 2.23 | 4.85 | I | 14 January 2021 | 30 | 15 January 2021 | 29 |

**Table 6.** *Cont.*

| H. Y. | M. | Sight Day | w | ARCV | DAZ | Pred. | First Day (1) | Days | First Day (2) | Days |
|---|---|---|---|---|---|---|---|---|---|---|
| 1442 | 7 | 12 February 2021 | 0.25 | 7.42 | 7 | P | 13 February 2021 | 29 | 13 February 2021 | 30 |
| | 8 | 13 March 2021 | 0.06 | 0.61 | 4.99 | I | 14 March 2021 | 30 | 15 March 2021 | 30 |
| | 9 | 12 April 2021 | 0.09 | 4.09 | 4.54 | I | 13 April 2021 | 30 | 14 April 2021 | 29 |
| | 10 | 12 May 2021 | 0.17 | 7.43 | 4.24 | P | 13 May 2021 | 29 | 13 May 2021 | 30 |
| | 11 | 10 June 2021 | 0.01 | 1.59 | 0.49 | I | 11 June 2021 | 30 | 12 June 2021 | 30 |
| | 12 | 10 July 2021 | 0.12 | 6.56 | 2.61 | I | 11 July 2021 | 29 | 12 July 2021 | 29 |
| 1443 | 1 | 8 August 2021 | 0.04 | 3.34 | 1.71 | I | 9 August 2021 | 30 | 10 August 2021 | 29 |
| | 2 | 7 September 2021 | 0.17 | 7.13 | 3.85 | P | 8 September 2021 | 29 | 8 September 2021 | 30 |
| | 3 | 6 October 2021 | 0.03 | 3.06 | 0.29 | I | 7 October 2021 | 30 | 8 October 2021 | 30 |
| | 4 | 5 November 2021 | 0.22 | 5.76 | 7.07 | I | 6 November 2021 | 30 | 7 November 2021 | 29 |
| | 5 | 4 December 2021 | 0.03 | 0.88 | 3.29 | I | 6 December 2021 | 29 | 6 December 2021 | 29 |
| | 6 | 3 January 2022 | 0.34 | 7.01 | 9.04 | P | 4 January 2022 | 29 | 4 January 2022 | 30 |
| | 7 | 1 February 2022 | 0.11 | 2.37 | 5.97 | I | 2 February 2022 | 30 | 3 February 2022 | 29 |
| | 8 | 3 March 2022 | 0.32 | 9.37 | 6.3 | P | 4 March 2022 | 29 | 4 March 2022 | 30 |
| | 9 | 1 April 2022 | 0.06 | 2.88 | 3.68 | I | 2 April 2022 | 30 | 3 April 2022 | 29 |
| | 10 | 1 May 2022 | 0.16 | 7.58 | 2.94 | P | 2 May 2022 | 30 | 2 May 2022 | 30 |
| | 11 | 30 May 2022 | 0.01 | 1.64 | 0.33 | I | 1 June 2022 | 29 | 1 June 2022 | 30 |
| | 12 | 29 June 2022 | 0.1 | 6.19 | 1.12 | I | 30 June 2022 | 30 | 1 July 2022 | 29 |

## 5. Discussion

Figure 5 gives a complete comprehensive analysis of the original observation dataset. First, the crescent moon would not be visible by any means available (except the CCD imaging) if the crescent width was less or equal to $0.12'$ ($ARCL$ is about $6.86°$ or elongation of $7.12°$), which proved Danjon's limit and Schaefer investigations. In the region of the lower limit of naked eye visibility, the moon would be certainly visible for any condition of the sky if the crescent width was $0.3'$ ($ARCL$ is about $11°$ or elongation of $11.25°$ provided that $DAZ = 0$), which confirmed the results of Maunder and Ilyas as the minimum elongation for naked-eye visibility. The crescent width might be greater than $0.3'$ and could not be seen by naked eye if its $DAZ$ was very large. For larger values of the crescent width $w > 1'$ ($ARCL > 20°$), it is obvious that the moon could not be seen by any means available if the $ARCV < 4°$, which confirms the finding of Ilyas. Furthermore, the minimum value of arc of vision for the crescent moon visibility is when $ARCV = 7°$.

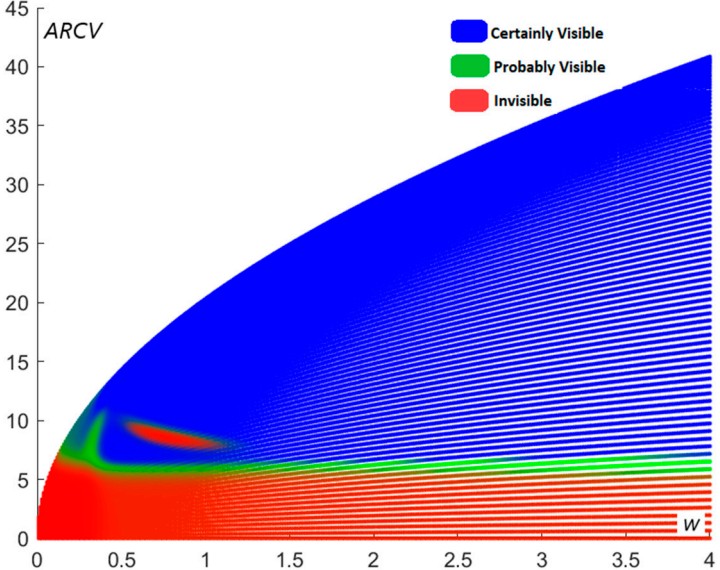

**Figure 5.** Performance of the Pattern Recognizer ANN for a test data.

Secondly, it is enough to divide the moon sight area into three regions rather than four as in Odeh's criterion or five as in Yallop's criterion. The invisible region is contiguous, and so is the visible one. The middle area was only a thin line between the two others.

Thirdly, there was a red spot inside the blue region. The reason was performing the old observations using the naked eye only. This spot should be green (probably visible) rather than red. Schmidt recorded these sightings as "invisible" rather than "probably visible" because he used the unaided sight only.

The crescent moon may be visible by the unaided eye in the green region, but it remains unlikely to be seen compared with the blue one. Furthermore, a visible area under an invisible area implies an issue related with recording sight results. The green region might contain blue spots, which indicates that the crescent could be visible by the unaided eye if the conditions were appropriate. Whenever the moon sight data approaches the red region, moon sighting will be less likely.

Figure 5 shows that the official Hijri Calendar in Iraq started the new lunar month on the following day of the moon sight, regardless of the crescent moon sight result (invisible or probably visible), yet most of the results were invisible. So one should consider that issue and reform the official Hijri calendar to use observational results rather than astronomical calculations, which most countries in the Middle East have used.

## 6. Conclusions

Many theories of moon sight attempted to simulate the problem of early visibility. All these theories were empirical and depended totally on the observational results to draw a line among the moon sight regions. This research suggested an alternative approach to visualize those regions depending on machine learning techniques. Deep-learning tools such as ANN's are helpful in simulating the moon sighting dataset.

The pattern recognizer ANN successfully simulated the moon sighting problem using observational data. The ANN verified the actual observational result for eight observations out of eleven. However, it is preferable to enhance the performance of the ANN using additional datasets. These results encourage collecting more data and training the ANN for improved accuracy, which helps construct robust Hijri calendars in any state or country. Unfortunately, the official Hijri calendar in Iraq does not depend on the observations of the astronomical societies and compromises their mathematical calculations.

These results open the way to propose a new criterion for crescent moon visibility which predicts the moon sight more likely. The results of this study will be the ground to create that criterion.

**Supplementary Materials:** The following supporting information can be downloaded at: https://www.mdpi.com/article/10.3390/computation10100186/s1. Table S1: The Complete Dataset of the Moon sight visibility.

**Funding:** This research received no external funding.

**Data Availability Statement:** The datasets used in this study are available in [16,17,21].

**Acknowledgments:** The author would express his deepest thanks and gratitude to his colleague Mohammed Shawkat Odeh the head of ICOP, for providing him with the essential resources and references throughout the period for preparing this study.

**Conflicts of Interest:** The author declare no conflict of interest.

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
