# Peer review of "A Pattern-Recognizer Artificial Neural Network for the Prediction of New Crescent Visibility in Iraq"

_computation, doi:10.3390/computation10100186_

Round 1
Reviewer 1 Report
Brief summary
The paper suggests the use of a shallow neural network to determine the visibility of the moon at the beginning of a lunar month. This information is then used for improving the official lunar calendars. The topic presented in the manuscript can be of interest to the readers of Computation; however, the paper is a little convoluted for an audience of non-astronomers.
General concept comments
The concept of using neural networks in astronomy is not novel; however, the application provided in this paper is quite fascinating on its own. As reported by the author, the computation of the lunar-month length has occupied human minds for millennia; and I personally believe that this might be the first time where a neural network model has been proposed to improve the current state-of-the-art approaches.
Although, I did not find any major issue with the proposed model, I was surprised to see the adoption of a softmax layer with three neurons in output. I initially thought to be a binary classification problem, but the author correctly implemented the model to fit the data in their original format. Please, consider to mention the type of response variable (e.g. categorical with three classes) somewhere in the introduction (and the abstract, if possible).
Specific comments
I was expecting a summary of the structure of the paper at the end of the introduction. Please, consider to include this as one paragraph at the end of Section 1.
In Section 2, I have seen different definitions of q in the formulas (2), (3) and (4). However, a subscript could help the reader to associate the different proposals from the literature.
Line 193. More information is needed to understand how to chose between "the two cases" (i.e., if the month has 29 or 30 days).
Line 229. It is not clear to me how the number of the neurons in the hidden layers has been chosen. There is no information on this in the manuscript.
Line 239. The data used are "observation data", but the words "generated data" appear in the text. This is unclear. Please, consider rephrasing if this is a reference to the aspect that the inputs of the neural network needs to be computed from other measurements (which are indeed observed).
Author Response
Dear reviewer,
Thank you for the review. I will consider all your comments an I will reply on them shortly.

Reviewer 2 Report
A very interesting application for a well known ANN algorithm. Authors do a good job with the abstract and introduction. The literature review on the area is well done and is an interesting read. However some deficiencies are visible in the following sections. Please address the following concerns for the next version of the paper:
1.) Line 14, Abstract: It is good to include metrics in the abstract, but the word "ratio" doesn't make sense. Did the authors mean "accuracy" ? Please correct
2.) Lines 81 and 82 : A physical explanation for the term "elongation" is important to better understand what the parameter means. Why is the shortest elongation metric important here ? Please clarify for the readers.
3.) Lines 146: Please re-write the sentence. Does it mean the "time taken by the moon to appear above the horizon after sunset ? "
4.) Please refer to "Table 2" in text.
5.) Figure 2: A better diagram can be created where the actual no of nodes in the NN layers is visible, Then this diagram can be used as a part(b) to explain the specifics.
6.) Line 229 : Also mention the input parameters here. If possible, please include them in the updated figure as well.
7.) Figure 3: Please add a legend to classify the different types of colored data.
8.) Figure 4: These very important figures are barely legible. Please use the wasted white space to make the written fonts as well as the figures legible.
9.) Line 280: Please explain each component of the result in details. The validation confusion matrix shows NAN values. No mention of a validation matrix is seen anywhere else in the paper. If validation is not relevant, please remove the val confusion matrix,.
10.) Fig 5: What do the colors indicate here ? Please provide a legend.
11.) Why do "Discussion" and "conclusion" have the same section numbers ?
12.) Some key aspects of the NN architecture description are missing. No mention of a hyperparameter optimization is present. It is important to justify that the best known architecture is used for the study. No mention of learning rate is present.
A great deal of literature search has been performed for the application aspect, but very little background info is present for the state of the art in terms of classification algorithms/pattern recognition.
13.) Conclusion section : A conclusion section should re-tell the story of the paper in a brief manner, with more emphasis on the results. Please re-write accordingly. Some of the statements seem vague without any numerical metrics
Author Response

(The authors gave the same response as above.)

Round 2
Reviewer 2 Report
Accepted with the corrections.